# Novel Bile Salt Stabilized Vesicles-Mediated Effective Topical Delivery of Diclofenac Sodium: A New Therapeutic Approach for Pain and Inflammation

**DOI:** 10.3390/ph15091106

**Published:** 2022-09-05

**Authors:** Tamer M. Mahmoud, Mohamed M. Nafady, Hanan O. Farouk, Dina M. Mahmoud, Yasmin M. Ahmed, Randa Mohammed Zaki, Doaa S. Hamad

**Affiliations:** 1Department of Pharmaceutics and Drug Manufacturing, Faculty of Pharmacy, MTI University, Cairo 12055, Egypt; 2Department of Pharmaceutics, Faculty of Pharmacy, Nahda University Beni-Suef, Beni-Suef 62764, Egypt; 3Department of Pharmacology and Toxicology, Faculty of Pharmacy, Nahda University Beni-Suef, Beni-Suef 62764, Egypt; 4Department of Pharmaceutics, College of Pharmacy, Prince Sattam Bin Abdulaziz University, Al-Kharj 11942, Saudi Arabia; 5Department of Pharmaceutics and Industrial Pharmacy, Faculty of Pharmacy, Beni-Suef University, Beni-Suef 62514, Egypt

**Keywords:** NSAIDs, transdermal drug delivery, bilosomes, permeation study, rat paw edema, the nuclear factor erythroid 2–related factor 2

## Abstract

The oral delivery of diclofenac sodium (DNa), a non-steroidal analgesic, anti-inflammatory drug, is associated with various gastrointestinal side effects. The aim of the research was to appraise the potential of transdermal delivery of DNa using bilosomes as a vesicular carrier (BSVC) in inflamed paw edema. DNa-BSVCs were elaborated using a thin-film hydration technique and optimized using a 3^1^.2^2^ multilevel categoric design with Design Expert^®^ software 10 software (Stat-Ease, Inc., Minneapolis, MI, USA). The effect of formulation variables on the physicochemical properties of BSVC, as well as the optimal formulation selection, was investigated. The BSVCs were evaluated for various parameters including entrapment efficiency (EE%), vesicle size (VS), zeta potential (ZP) and permeation studies. The optimized BSVC was characterized for in vitro release, Fourier transform infrared spectroscopy (FTIR), transmission electron microscopy (TEM) and incorporated into hydrogel base. The optimized DNa-BSVC gel effectiveness was assessed in vivo using carrageenan-induced paw edema animal model via cyclooxygenase 2 (COX-2), interleukin 6 (IL-6), Hemooxygenase 1 (HO-1) and nuclear factor-erythroid factor2-related factor 2 (Nfr-2) that potentiate anti-inflammatory and anti-oxidant activity coupled with histopathological investigation. The resulting vesicles presented VS from 120.4 ± 0.65 to 780.4 ± 0.99 nm, EE% from 61.7 ± 3.44 to 93.2 ± 2.21%, ZP from −23.8 ± 2.65 to −82.1 ± 12.63 mV and permeation from 582.9 ± 32.14 to 1350.2 ± 45.41 µg/cm^2^. The optimized BSVCs were nano-scaled spherical vesicles with non-overlapped bands of their constituents in the FTIR. Optimized formulation has superior skin permeability ex vivo approximately 2.5 times greater than DNa solution. Furthermore, histological investigation discovered that the formed BSVC had no skin irritating properties. It was found that DNa-BSVC gel suppressed changes in oxidative inflammatory mediators (COX-2), IL-6 and consequently enhanced Nrf2 and HO-1 levels. Moreover, reduction of percent of paw edema by about three-folds confirmed histopathological alterations. The results revealed that the optimized DNa-BSVC could be a promising transdermal drug delivery system to boost anti-inflammatory efficacy of DNa by enhancing the skin permeation of DNa and suppressing the inflammation of rat paw edema.

## 1. Introduction

Inflammation is a normal physiological body defense mechanism against stimuli, cellular stress, pathogens, toxins and cellular injury [1]. Generally, an acute inflammatory reaction is the primary cause of a chronic inflammatory response and subsequently the development of several diseases that may induce cell death or organ damage [2,3].

The excessive proinflammatory and inflammatory cytokine liberation is a primary cause of developing acute tissue inflammation [4]. In a rat model, the carrageenan-induced paw edema is an acute inflammatory that exploits the effect of the novel anti-inflammatory drug delivery models [5]. Additionally, it is implicated in the startup inflammatory process introducing via prescience proinflammatory cytokines Interleukins 6 (IL 6), tumor necrosis factor-α (TNF-α) and cyclooxygenase (COX 2) where it plays a key role in inflammation [6]. Consequently, evidence suggests that heme oxygenase-1 (HO-1) is a crucial component of the anti-inflammatory processes of the inflamed cells under the control of nuclear factor erythroid 2-related factor 2 (Nrf2) that preclude inflammation from worsening [7].

As a nonsteroidal anti-inflammatory drug (NSAID), diclofenac sodium (DNa) has been utilized to manage chronic and acute diseases connected with inflammatory conditions, exclusively those including the musculoskeletal system. It is believed to exert efficient analgesic and anti-inflammatory actions by inhibiting cyclooxygenase-2 specifically (COX-2) [8]. Although NSAIDs are effective when taken orally, many individuals have problems with tolerance that include gastrointestinal distress and significant cardiovascular and renal side effects [9].

An unusual route to advance the tolerability and safety profile of NSAIDs has been the implementation of topical formulations designed to deliver the drug transdermally directly to underlying tissues while minimizing systemic exposure [10]. In transdermal drug delivery systems, the stratum corneum (SC), which has the largest barrier function, reduces the amount of drugs absorbed in the skin layers [11]. It is critical to make sure that a sufficient number of active medications enter and stay in the skin. Furthermore, sufficient amounts of active medication at specific places on the skin help to improve clinical effectiveness in the treatment of inflammation. As a consequence, numerous ways have been developed in recent years to decrease the skin barrier and improve medication absorption via the skin, such as chemical and physical augmentation methods containing magnetophoresis, iontophoresis, electroporation and microneedles [12]. However, their uses are limited due to therapeutic practicality and toxicity.

The development of transdermal administration systems with permeation enhancers is one more method to boost drug penetration through the skin [13]. Essentially, positive results have been realized utilizing the lipid-based vesicles as penetration enhancers for both hydrophobic and lipophobic substances via the skin [14]. Vesicular systems have shown promise in attenuating the skin’s SC intercellular lipid barrier [15], thereby, permitting drug diffusion beyond the skin layer, followed by systemic absorption through superficial dermal vessels. Bilosomes as vesicular carriers (BSVC) are nanostructured lipid carriers made up primarily of amphiphilic bile salts that show patent penetration drug capability through intact biological membranes including skin and intestine [13]. Furthermore, transdermal delivery is heightened due to the high flexibility of BSVC, which improves the penetration through the SC and skin-deep layers [16,17]. Moreover, compared to typical liposomes, the addition of bile salts, such as sodium deoxycholate (SDC), improves the system’s colloidal stability [18]. It also has a nanoscale diameter and a fluidizing action, which boosts transdermal delivery [19,20]. BSVCs have been employed in several studies to enhance the transdermal delivery of different drugs such as lornoxicam [19], tizanidine hydrochloride [21] and ondansetron hydrochloride [22].

Hydrogels are three-dimensional networks of hydrophilic polymers, capable of absorbing large volumes of biological fluids or water [23]. Because of their unique physical properties, such as biodegradability, flexibility, high porosity, biocompatibility and controlled drug release, they are fascinating vehicles for drug delivery applications [24]. Cellular encapsulation, hydrogel microparticles, hydrogel sandwich systems, porous hydrogels, fibrous hydrogel scaffolds, bio-printed scaffolds, microfluidics and microwells are just a few of the hydrogel platforms that can be used in functional tissue models [24]. Drugs may be trapped in hydrogels due to their porosity and released at a frequency determined by the molecule diffusion coefficient across the hydrogel network [25]. Furthermore, they have no effect on metabolic processes, and drugs can pass through the hydrogels without difficulty [26].

To date, no documented studies have appraised the prospective of BSVC as transporters for DNa. Therefore, the work in this manuscript was concerned in three major points: firstly, to tailor DNa-BSVC using a multilevel categoric design to scrutinize its aptitude to increase the transdermal permeation of DNa and then handpick the optimized formulation accompanied by a high desirability index. Secondly, we want to conduct ex vivo and in vitro testing of the optimized formulation to evaluate its safety and suitability for transdermal administration. Finally, the study evaluates the effect of DNa-BSVC in carrageenan-induced paw edema. Then, the results were compared with oral DNa solution and transdermal free DNa in a gel. This study could provide a promising transdermal drug delivery system to boost the anti-inflammatory efficacy of DNa by enhancing the skin permeation of DNa, and suppressing the inflammation of rat paw edema.

## 2. Results and Discussion

### 2.1. Experimental Design and Statistical Analysis

As illustrated in Table 1, it was observed that independent factors had a significant effect on all responses. Predicted R^2^ is usually less than the adjusted R^2^, i.e., the difference is less than 0.2. “Adeq Precision” is the ratio between a significant noise ratio that should be greater than four to enable the model to navigate the design space.

Table 1 shows that all responses depicted a ratio greater than four therefore; this model can navigate through the design space to declare the significant effect of independent factors on the responses under investigation [27,28].

The models accurately described the variability of the observed data, as seen by the variability of the observed data (Table 1). Figure 1A–C The model diagnostic graphs of the three causal factors which reveal that the data is well-fitting, with no identifiable patterns of residual errors that largely follow a normal distribution. The higher and lower confidence intervals (C.I.) of lambda, the present lambda = 1, are determined using Box–Cox for power transformation (Figure 2), which provides the higher and lower confidence intervals (C.I.) of lambda. As a result, the original data were used to conduct the statistical analysis.

### 2.2. DNa-BSVCs Characterization

#### 2.2.1. Effect of Independent Variables on EE%

As illustrated in Table 2, DNa entrapment was in the range of 61.7–93.2% for all vesicles. The combined effect of the two independent variables (span type and bile salt type) on the EE% of DNa-BSVCs at the lowest levels of the third variables (bile salt concentration) is demonstrated in (Figure 3A). The significant impact of independent factors on the EE% of DNa-BSVCs was shown in (Table 1). 

The EE% values of Span 60-contained BSVCs formulations were superior to those of Span 40, which might be succumbed to the higher phase transition temperature of Span 60 that offers greater drug encapsulation [29]. Furthermore, the longer saturated alkyl chain of Span 60 (C14), the more the BSVC bilayer stable with subsequent increase in EE% compared with Span 40-contained BSVCs formulations [30]. Additionally, the lower HLB value of Span 60 (4.7) compared to Span 40 (HLB 6.7), which makes it more hydrophobic and results in better EE% [31]. Similar results were obtained by [32] who reported that the lower the HLB of the used surfactant, the higher the entrapment efficiency of colchicine within the prepared noisome.

Similarly, SDC-based BSVCs formulations exhibited significant larger EE% than those combined SC and SGC, *p* < 0.05. A notable explanation may be the surface active property and capacity to integrate into the surface of the bilayer membrane which in turn increases the bilayer membrane elasticity and drug solubility into the membrane and directly affects the EE% [33]. Moreover, BSVCs fabricated using SDC with the lower HLB values (16) revealed larger EE% compared with BSVCs including SC and SGC possessing the higher HLB values (18) and (23.1), respectively, using the same amount of DNa to load the BSVCs. Therefore, the EE% values of BSVCs increased as HLB of bile salt decreased (Ashraf et al., 2018). Salem et al. [34] reported comparable results, stating that the higher the HLB of the bile salt, the lower the EE%.

Although the bile salt had no influence on entrapment (*p* = 0.739), several studies have showed that entrapment of BSVCs decreased as bile salt concentration increased. These observations could be attributed to the fluidizing effect of bile salt on the bilayer membrane of vesicles, which causes drug leakage [17]. Furthermore, at high concentrations, the bile salt may form mixed micelles, improving the medication’s solubility in the dispersion medium and decreasing trapping [35].

#### 2.2.2. Effect of Independent Variables on VS

As depicted in Table 2, the mean vesicle size of the BSVCs fluctuated from 120.2 ± 0.65 nm to 780.4 ± 0.99 nm. Typically, bile salt-containing vesicles are spherical with a low potential for agglomeration and, consequently, have modest particle sizes [36]. All formulations of BSVCs had PDI values between 0.31 and 0.62, which may be considered an acceptable mid-range suggesting a fair size distribution and assuming formulation uniformity [37].

Our findings for span type revealed that the analyzed variables had a significant influence on BSVC mean size (*p* < 0.05). The average size of Span 60-based BSVCs was larger than that of Span 40-based vesicles, as shown in (Figure 3B). Presumably, the lower the HLB of the surfactant, the smaller the vesicle size produced. Thus, Span 40-based BSVCs (HLB 4.7) were smaller sized than those containing Span 60 (HLB 6.7). The observed link between the surfactant HLB and vesicle size can be attributed to a decrease in surface energy caused by an increase in hydrophobicity, resulting in the creation of smaller vesicles [38]. Yoshioka et al. [39] revealed comparable findings on the estimation of the properties of several spans containing niosomes. In addition, increasing the hydrophilicity of the surfactant led to more water being taken in, which made the vesicles bigger [40]. Moreover, increasing the lipophilic affinity of the surfactant makes the droplets smaller and slows down the transfer of mass within the droplets, which slows the growth of a nucleus and makes the VS smaller [41]. 

Regarding the type of bile salt, BSVCs prepared using SGC were significantly higher than those prepared using SC and SDC. This could be due to the higher ZP values of these nano-vesicles formulated with SGC compared to other formulations. The higher ZP increased the repulsion force between the charged adjacent vesicular bilayers resulting in an increase in space among them, leading to the formation of relatively larger vesicles [42]. These findings are in line with those of an earlier study [43].

Bile salt impacts the VS by lowering surface tension and interfacial tension between vesicle bilayers, thereby reducing the space between vesicle bilayers [33,44]. However, increasing the amount of bile salt in a high concentration causes it to tend to aggregate itself [33]. The anionic nature of bile salts and their steroid structure increase VS in two ways: the first is by growing the internal aqueous core space, and the second is by increasing the steric repulsive force between bilosome bilayers, which may increase bilosome bulkiness [19]. In contrast to previous research that found a decrease in VS with increasing bile salt content, the present findings show that bile salt has no effect on the curvature of the vesicle membrane [45].

#### 2.2.3. Effect of Independent Variables on ZP

As illustrated in Table 2, Appendix A, the ZP of formulated BSVCs ranged between (−) 23.8± 2.65 and (−) 82.1 ± 12.63 mV. Although the span type had an insignificant effect on zeta potential (Figure 3C), some reports have shown that the zeta potential of BSVCs increased when Span 40 was used compared to Span 60. The decrease in the surfactant’s surface free energy and decreasing the surfactant’s lipophilicity which increase zeta potential, may be responsible for these findings [46]. 

Bile salts are electrostatic stabilizers that boost surface charge density to the vesicles, making them more stable. Furthermore, owing to the negative charge of the bile salt, a high bile salt level is followed by an increase in the ZP negative charge of the produced vesicles [47]. These findings are backed up by prior research in the literature [48].

According to statistical analysis of the ZP value, DNa-BSVCs were increased by increasing the bile salt concentration. These findings were unexpected and could be due to the anionic nature of bile salt, as it would be predicted that increasing its concentration would increase the negative of the produced vesicles [27]. Ref. [49] shared similar results in their investigation on the formulation of a curcumin nano-bilosomes system.

#### 2.2.4. Effect of Independent Variables on Ex Vivo Skin Permeation Study

Regarding the use of BSVCs as a transdermal platform, ex vivo diffusion experiments were analyzed to predict their efficacy in vivo. Ex vivo skin permeation profiles of DNa-loaded BSVCs via rat skin relative to DNa solution are represented in Appendix A. The developed BSVCs demonstrated significantly higher skin permeation compared with a drug solution comprising an equal quantity of DNa (*p* < 0.05). From Appendix A, it is noticed that only 364.6 ± 37.00 µg/cm^2^ of DNa solution was permeated via rat skin over 21 h; meanwhile, the cumulative amount of DNa permeated from BSVCs ranged from 582.9 ± 32.14 to 1350.2 ± 45.41 µg/cm^2^ Table 2. Collectively, the calculated permeation parameters of the inspected DNa-loaded BSVCs via rat skin are summarized in Table 3. The investigated BSVC exhibited transdermal flux values in the range of 31.97 ± 0.98 to 87.24 ± 0.356 µg/cm^2^ h versus 23.44 ± 0.743 µg/cm^2^ h for the control DNa solution. Therefore, the obtained results emphasized the preponderance of BSVCs in both sustaining DNa release and promoting its skin diffusion from 1.6- to 3.7-fold greater than the DNa solution.

Figure 3 D depicts the effect of formulation factors on the Q21 of DNa-loaded BSVCs. All factors produced a positive regression coefficient, indicating that increasing their levels had a favourable impact on transdermal diffusion. This increased transdermal DNa delivery from the produced BSVCs could be attributed to the vesicular carrier’s ability to offer DNa as a minute colloidal dispersion with a greater surface area, hence shortening the DNa diffusional path length across the skin [50]. Furthermore, modifying BSVCs with EAs may improve vesicular penetration into skin pores by enhancing vesicular deformability [31]. When applied under non-occlusive settings, increasing vesicular flexibility increases its propensity to hold and bind water, which improves vesicular integration into SC lipid to ensure optimal hydration conditions [51]. Furthermore, bile salts penetrate the intercellular lipids, interact aggressively with keratin filaments in corneocytes, and therefore open up the dense and ordered keratin structure, altering its barrier characteristics significantly [52]. Furthermore, bile salts increase DNa penetration via both the paracellular and transcellular routes. This could be attributed to the ability of bile salts to extract calcium and multivalent ions, which would improve protein accessibility in the stratum corneum’s lower portion. Binding bile salt molecules to highly accessible proteins causes them to denature, resulting in stratum corneum enlargement and corneocyte cohesion loss [53]. When compared to a DNa solution, all of the aforementioned conditions boosted transdermal distribution of DNa via BSVCs. In our investigation, BSVCs with Span 60 were more easily penetrated than BSVCs with Span 40. The observed variation in skin permeation parameters could be attributed to particle size differences in the developed BSVCs. This finding is consistent with the findings of El Menshawe et al. [54], who observed an increase in Q21 percent with decreasing particle size following the formulation of spanlastic for fluvastatin administration.

A plausible finding is that bile salt with concentration 18-decorated vesicles yielded significantly higher permeable rates compared to bile salt with concentration 8-decorated vesicles. This could be attributed to the fact that the higher EA concentrations lead to increased fluidity of the vesicles membranes and increase the permeability of the vesicles membranes and thus promote the drug release [31]. These suggestions were supported by Almahallawi et al. [55], who found that using increased concentrations of sodium cholate as EA led to a significant increase in ciprofloxacin released per time from transferosomes.

#### 2.2.5. Formulation Optimization

In order to create a high-quality pharmaceutical product that met the study’s objectives, the best formula with the optimal values of dependent variables was determined through the optimization procedure. A numerical optimization method based on the desirability function was used in this work to overcome various and conflicting responses. The suggested optimized DNa-BSVC (S12) showed the highest desirability of 0.889. Optimized DNa-BSVC (S12) created with (Span 60) and 18 mg SDC was determined to meet optimization criteria (minimum values for VS and ZP, maximum values of EE%, % Q21 h). This optimized formulation showed VS of 275.33 nm, EE% of 95.41%, ZP of −86.32 mV and % Q21 h of 1273.85 µg/cm^2^. Table 4 shows the predicted and observed dependent variables of the optimized S12 to reinforce the optimization process validity. Obviously, a high similarity between the observed and predicted values was depicted; therefore, S12 was selected for further investigation.

### 2.3. Optimized DNa-BSVC Characterization

#### 2.3.1. In Vitro Release Evaluation 

The in vitro release profiles depicted in Figure 4 show the cumulative amount of released DNa from drug solution and the prepared BSVC as a function of time. The DNa solution released 99.55 percent of the collected amount in 3 h, but the DNa-BSVC solution released only 36.48 percent. As a result, the rate at which DNa was released from DNa-BSVC was slower than the rate at which DNa was released from the comparable DNa solution, demonstrating that the bilosomal system might delay DNa release.

It is possible that the potential of BSVC to act as a drug reservoir may be responsible for the slow continuous release of DNa from different formulations over 8 h [56]. A biphasic pattern was seen in the drug-release profiles of the medication DNa from BSVC (% Q3 h, 31.68%), with a high initial burst release in the first three hours (% Q3 h, 31.68%) and a slower release phase. Detachment from the vesicle’s exterior surface may be responsible for this initial surge, followed by a longer period of time when DNa gradually partitions out of its bilayer into the releasing media [57]. For all prepared BSVs, DNa’s biphasic release characteristics are likely to be of tremendous advantage. An anti-inflammatory drug’s rapid initial release, followed by a gradual release over the next eight hours, would allow the patient to remain under therapy all day long with fewer doses. In keeping with previous studies [58], these findings support the previous ones.

#### 2.3.2. Morphological Evaluation

TEM was assessed to investigate the morphology of the optimized formula (S12), as shown in Figure 5. The TEM micrograph represented spherically shaped, non-structured, unilamellar, well dispersed, well separated vesicles. The regular spherical vesicles obtained may be attributed to the enhanced flexibility and the reduced surface tension [59]. It is noteworthy that the obtained size by TEM is lower than that size obtained from the dynamic light scattering technique (DLS) approach utilizing a Zetasizer NanoZS 7.11 (Malvern Instruments, Malvern, UK) because the analytical principle used was different in both techniques. DLS is an intensity-based technique in which the size distribution result is the hydrodynamic size average of the nanoparticles and is frequently impacted by the presence of aggregates, large particles or dust [60]. Particularly, the nanovesicles evaluated using DLS techniques are in solution and surrounded by layers of the medium used, which increases their measurement size. However, microscopic examination by TEM relies heavily on nanoparticle tracking analysis (NTA), and the observation of a nanoparticle containing a droplet on a TEM grid is often performed after the routine drying method of air drying. Individual nanoparticles are tracked using NTA, a numerically based method (single-particle tracking) [60]. Thus, the latter can deliver a precise number-based average dimension with little bias for pure samples [61]. Therefore, TEM will achieve a smaller size than DLS. Although the procedures yielded differing vesicle sizes, S12 demonstrated good homogeneity reflected from the PDI value = 0.39 ± 0.021, indicating the narrow size distribution; thus, more stable and homogeneous formulations were created.

#### 2.3.3. Fourier Transform Infrared Spectroscopy Study (FTIR) of the Optimized DNa-BSVC Formulation

Based on comparisons between the optimized formulation fingerprint region (below 1500 cm^−1^) and its components (DNa, Span 60, SDC, lecithin and cholesterol), FTIR spectra confirmed BSVC production (Figure 6). DNa observed distinctive peaks at 3385 cm^−1^ (NH stretching of the secondary amine), 1574 cm^−1^ (-C=O stretching of the carbonyl ion), 1557 cm^−1^ (C=C ring stretching) and at 746 (C-Cl stretching). The O-H group was found at wavenumber 3330 cm^−1^, alkanes at wavenumber 1449 cm^−1^ and aromatic rings at wavenumber 1449 cm^−1^ in Span 60 spectra. The characteristic bands of SDC are shown at wavenumber 2939 cm^−1^ (aliphatic C-H) and wavenumber 1562 cm^−1^ (COO−). Lecithin showed characteristic peaks at 2919 cm^−1^ and 2921 cm^−1^ (due to stretching of −CH of methylene group) and at 1739 cm^−1^ and 1737 cm^−1^ (due to C=O group stretching). Cholesterol O-H group appears at wavenumber 3341 cm^−1^ and alkanes, aromatic rings at wavenumber 1454 cm^−1^. FTIR spectra of DNa and the optimized formulation showed no change in their function group areas, indicating that the drug and other formulation components do not interact chemically. The fingerprint regions, on the other hand, are not overlaid, indicating that physical character changes have occurred as shown in Figure 6.

### 2.4. Characterization of DNA-BSVC Hydrogels

#### 2.4.1. Rheological Characterization of the Hydrogel Formulations

Table 5 shows the rheological properties of the prepared hydrogel formulations. The apparent viscosities were determined at maximum and minimum shear rates. By studying the effect of mucoadhesives used (Carbopol 971P) on viscosity of DNa hydrogel and DNa-BSVC hydrogel, we can deduce that incorporation of mucoadhesives increased the viscosity of the formulations. These results were in agreement with the previous literature [62]. Complete rheograms were generated by graphing the shear rates against shear stresses. The figures represent a counterclockwise hysteresis curve, namely hysteresis loop. DNa hydrogel and DNa-BSVC hydrogel showed thixotropy with shear rate thinning. Figure 7 plots the results of viscosity measurements for each hydrogel at various shear rates. It should be noted that there was an opposite relationship between the shear rate and the viscosity. So, a typical shear rate thinning behavior was obtained.

Thixotropy means that the viscosity of the fluid reduces with the rise of shear stress. After the shear stress is withdrawn, the viscosity returns gradually to the former state under isothermal conditions. The characteristic thixotropy flow curve descends moving toward the left compared with the ascending curve. Needless to say, thixotropy is a desirable character in pharmaceutical gels. The degree of thixotropy was calculated quantitatively by determining the area of hysteresis loop formed between the up and the down curves of the rheograms using a trapezoidal rule. In addition, the flow activity of the hydrogel formulations prepared was analyzed using Farrow’s equation.

DNa-BSVC hydrogel have favorable thixotropy and viscosity, so the overriding concern in developing a mucoadhesive hydrogel to extend the residence time of the drug formulation in stratum corneum could be achieved.

#### 2.4.2. Ex Vivo Permeation Study 

With regard to the application of DNa as a nano-transdermal form, studies on ex vivo permeation were investigated to predict their performance in vivo. Ex vivo skin diffusion outlines of DNa-BSVC gel via rat skin comparative to DNa gel were embodied in Figure 8. The developed BSVC hydrogel demonstrated significantly higher skin permeation paralleled with DNa hydrogel involving an equivalent amount of DNa (*p* < 0.05). From Figure 8, it is noticed that 337.55 ± 51.25 μg/cm^2^ of DNa hydrogel was permeated through rat skin over 21 h; meanwhile, the cumulative amount of DNa permeated from BSVC hydrogel was 995.23 ± 56.97 μg/cm^2^. The investigated BSVC hydrogel exhibited transdermal flux values of 65.37 ± 9.12 μg/cm^2^ h versus 23.26 ± 4.84 μg/cm^2^ h for the DNa hydrogel. Consequently, the results obtained highlight the predominance of DNa-BSVC hydrogel maintaining DNa release and facilitating skin diffusion 2.5-fold higher than DNa hydrogel. It has been demonstrated that the DNa-BSVC hydrogel possesses the maximum diffusion. These findings can be explained by the fact that Carbopol 971P has a very high percentage (58–68%) of carboxyl groups forming hydrogen bonds with sugar residues in an oligosaccharide chain of mucin glycoprotein in epithelial tissues, resulting in a reinforced network between polymer and epithelial tissue of skin [63] leading to extended retention and increased permeation through epithelial tissues [63]. Secondly, it is reported that the anionic polymer Carbopol 971P has permeation-enhancing capabilities since it can bind Ca^+2^ present in the epithelial tissue [64]. Thirdly, Carbopol 971P’s polymer chain undergoes decoiling due to electrostatic repulsion between its ionized carboxyl groups. Thus, Carbopol 971P swells due to the epithelial tissue’s absorption of water, which causes intimate penetration into the skin, hence localizes the formulation in the skin and increasing the drug concentration gradient across the epithelium [65]. This results in increased drug solubility and diffusion from the hydrogels due to substantial swelling of the ionized Carbopol 971P [66].

### 2.5. Histopathological Study

After topical application of the optimized BSVC, skin segments were examined under a light microscope to guarantee safety. Group A’s epidermis had normal keratin coated layers. The skin appendages were normal; the dermis included connective tissue, and there was cutaneous vascularity (Figure 9a). The epidermis layer had a typical keratin squamous layer, and the dermis layer was devoid of edema, erythema and inflammation when hairless rats’ skin was treated with DNa-BSVC gel (Figure 9b).

### 2.6. In Vivo Pharmacological Study

#### 2.6.1. Rat Paw Volume % Inhibition

The carrageenan-induced paw edema model was used to evaluate the difference between DNa-based gel and DNa-BSVC gel treatments anti-inflammatory activity in the rat received carrageenan of a single dose 0.1 ml of 1.5% induced paw edema experiment, and the results are shown in Table 6. The treatment effect was tracked by observing the improvement of rat paw edema in response to drug administration. The carrageenan injection induced a significant increase in paw thickness to 0.9 cm after 1 h of injection and remained constant until 5 h. The DNa-BSVC gel significantly decreased paw edema to (50.9%), (59.6%), (67.9%), (67.9%) and (69.1%), respectively, after administration regarding the time interval 1 h to 5 h, respectively, compared to the positive control. The DNa-based gel group significantly decreased paw edema to (11.3%), (15.3%), (20.3%), (22.7%) and (23.2%) after administration regarding the time interval 1 h to 5 h, respectively, compared to the positive control. The results demonstrated that DNa-BSVC gel paw edema volume was significantly restored to normal in contrast with the DNa-based gel group that showed less improvement to the paw volume compared to normal control Table 6.

In agreement with our results, the intra-plantar injections of carrageenan can induce hyper-algesia and allodynia in rodents, allowing for the assessment of hyperplasia and pain response [67,68]. Ercan et al., 2013 [69] demonstrated that topical application of diclofenac significantly reduced pain and subsequent edema induced by carrageenan. Additionally, previous data revealed that the paw edema influenced by carrageenan reduced inflammation by receiving nanoemulgel diclofenac relative to conventional gel [70]. Furthermore, Anita et al., 2021 [71] reported that the diclofenac nanocarrier formula is convenient for reducing NSAIDs pharmacological adverse effects and improving the efficiency of local medication delivery to the inflamed joint that may be an excellent option for high-risk patient populations. Likewise, the neuropathic incisional pain in a patient model proved that the application of diclofenac nanogel on a patient scalp significantly decreases inflammation of the peripheral nerves and the supporting structures that can cause neuropathic pain [72]. Sakat et al., 2014 [73] confirm that diclofenac has a potent anti-inflammatory effect in reducing carrageenan-induced paw edema in experimental rats after 2–3 h in a dose-dependent manner.

#### 2.6.2. Cyclooxygenase 2 (COX-2) and Interleukine 6 (IL-6) Paw Tissue

The inhibitory effect of DNa-BSVC gel regarding COX-2 enzyme was evaluated in rats exposed to carrageenan induced significant increase in COX-2 levels to 305.91% at *p* > 0.05 compared to normal control rats. DNa-based gel administration revealed a 37.97% inhibition to the COX-2 levels compared to carrageenan, while DNa-BSVC gel significantly inhibits COX-2 levels to about 57.82% compared to the carrageenan group (Figure 10A).

Additionally, the results for IL-6 inflammatory cytokines revealed that rats subjected to carrageenan significantly increased its level to about 426.64% compared to the normal control group. DNa-based gel represents a minimal decrease in IL-6 levels to about 20% compared to the carrageenan group. In contrast, DNa-BSVC gel significantly decreased IL-6 levels to about 50.32% (Figure 10B). The results obtained for both COX-2 and IL-6 regarding DNa-BSVC gel show a significant anti-inflammatory effect compared to DNa-based gel.

The previous data reveals that during inflammatory processes, the overexpression of COX-1 and COX-2 plays a critical role in inflammation signaling pathway [74]. Cyclooxygenase was found to be distributed in many sites of the human body tissues and involved as a potential anti-inflammatory agent [75]. In vivo and in vitro study reveals a cross talk between the nitric oxide and prostaglandins; NO promotes an increase in the COX-2 half-life by generating free radicals, inhibiting COX-2 autoactivation [76]. Additionally, patient inflammatory conditions could be revealed via serum biochemical or histopathological estimation of inflammatory markers as CRP, interleukins, COX-2, NF-kB and TNF-α [77]. Moreover, Saleem et al., 2021 validate the DNa regulatory anti-arthritic, anti-inflammatory effect as standard treatment in rat CFA and a carrageenan-induced inflammatory model compared to herbal medicine as it regulates pro-inflammatory mediators [78]. Inflammation management can prevent excessive organ damage from an aggressive immune response or development of chronic disease [79].

It has been established that inflammatory cytokines such as IL-6 and IL-8 are involved in inflammatory processes by aggravating proinflammatory cells, angiogenesis and the induction of proliferation [80]. In agreement with our results, a collagen-induced rheumatoid arthritis revealed that the use of diclofenac as a standard anti-inflammatory standard drug decreased IL-6 levels [81]. Furthermore, dos Santos Haupenthal et al. [81] showed that diclofenac-linked gold nanoparticles enhance proinflammatory cytokines accelerate tissue repair. In addition, Gonçalves et al. [82] proved that diclofenac nanoparticles are an effective anti-inflammatory drug delivery system that induced a significant decrease in PGE2 and IL-6 concentration in LPS-activated macrophages, enhancing desirable properties to locally control macrophage response.

#### 2.6.3. Nuclear Factor Erythroid 2 (Nrf-2) and Hemoxygenase-1 (HO-1) Serum Levels

Based on the abovementioned results, the DNa-based gel and DNa-BSVC gel investigated had significant impact on the systemic inflammatory response considering that serum Nrf-2 levels were found to be lowered in carrageenan positive control rats to about 74.83% compared to normal control rats. Furthermore, the Nrf-2 levels measured in the DNa-based gel significantly increased Nrf-2 serum levels to 2.24% compared to the carrageenan positive control group, while the DNa-BSVC gel represents a significant increase in Nrf-2 levels to 1.67% compared to the carrageenan positive control group with no significant difference with DNa-based gel (Figure 11A).

Additionally, the HO-1 activity assessed showed that rats subjected to carrageenan significantly decreased its level to 78.39% compared to normal control rats. However, treatment with both DNa-based gel and DNa-BSVC gel significantly increased its level to 2.98% and 1.82%, respectively, compared to the carrageenan positive control group. In addition, DNa-based gel showed a significant enhancement to HO-1 levels rather than DNa-BSVC gel (Figure 11B).

A previous study revealed that there are certain bioactive proteins that attenuate wound-healing by activating the Nrf2/HO-1 signaling cascade [83]. However, another study showed that HO-1 is a key regulator downstream gene of the Nrf2 signaling pathway that plays a key regulatory role against oxidative damage and inflammation [84]. In agreement with previous data, injection with carrageenan induces key inflammatory mediators release including HO-1 and Nrf-2 [84]. Previous research proved that Nfr-2 plays a key regulatory role in the elimination of ROS, protein detoxification and reduction of protein oxidation [85]. The Nfr-2 binds to the Maf protein, developing a heterodimer. This complex dimer abundantly binds to the antioxidant response element (ARE) or Maf recognition elements (MAREs) that make it play an important role in the cellular defense antioxidant pathway. The Nfr-2-Maf dimer serves as HO-1 promoter [86,87]. Another study agrees with our results that activation of the Nrf2/HO-1 pathway suppresses the production of inflammatory molecules such as TNF and IL-6 [88,89,90]. Therefore, along with previous studies, our data presented here suggest that DNa-BSVC gel inhibits HO-1-induced inflammatory pain by activation of Nrf2 and HO-1, suggesting an antioxidant and anti-inflammatory effect of our formula.

#### 2.6.4. Histopathological Study

##### Microscopic Examination (Routine H& E Staining)

Figure 12 shows a images of paws from the different groups (Figure 12a1,b1,c1,d1). Paws from the control group (Figure 12a1) revealed a normal image. In contrast, paws of animals from the positive control group (PC) (Figure 12b1) exhibited severe paw edema. Moderate improvement was noticed in the DNa-based gel groups (Figure 12c1), while DNa-BSVC gel-treated paws were significantly improved. (Figure 12d1).

Histopathological microscopic examination of paws from the control group (Figure 12a2,a3) revealed normal histology of bone, surrounding soft tissues and covering skin. In contrast, the paws of animals from the PC group (Figure 12b2,b3) exhibited marked edema in the periarticular and subcutaneous tissues surrounding the bone coupled with extensive hemorrhages were frequently detected in almost all examined sections. Moderate improvement was noticed in the DNa-based gel group (Figure 12c2,c3) in which mild to moderate edema in the soft tissues surrounding the bone was noticed. However, the best improvement was detected in the DNa-BSVC gel group (Figure 12d2,d3) as periarticular soft tissues were apparently normal and free from infiltration and aggregation of edema and inflammatory cells.

In agreement with our histopathological examination, Burayk et al. [91] revealed that rats subjected to carrageenan showed marked thickening of the dermal layer, inflammatory reaction in the deep dermis, and wide separation between fibers due to edema. In addition, a rheumatoid arthritis aggressive inflammatory model agrees with us that rats that received diclofinac presented a reduction in the number of inflammatory cells and hyperplasia of the synovium [92]. In addition, A transdermal formulation containing leflunomide and diclofenac-based gel showed significant relief of inflammation [93]. Likewise, Chang et al. [94] recently showed that haylarunic-liposome-loaded dexamethason-diclofinac applied to experimental osteoarthritic-induced mice locally reduced knee joint inflammation, edema and bone and cartilage damage.

## 3. Materials and Methods

### 3.1. Materials

Diclofenac sodium (DNa) was kindly provided by Misr Pharmaceutical Co. (Cairo, Egypt). Span 40 (sorbitan nonopalmitate) and Span 60 (sorbitan monostearate) were kindly supplied from CID Co. (Cairo, Egypt). Sodium cholate (SC), sodium deoxycholate (SDC), and sodium glycocholate (SGC) were purchased from Acros Organics (Cairo, Egypt). Cholesterol, lecithin and carrageenan (catalog number: C1013) were procured from Sigma-Aldrich (St. Louis, MO, USA). Methanol (HPLC grade) and methylene chloride (HPLC grade) were purchased El-NASR Pharmaceutical Chemicals Co. (Cairo, Egypt). Dialysis bags (Mol. Wt. cut off = 12,000 Da) were acquired from SERVA Electrophoresis GmbH (Heidelberg, Germany). Hemooxygenase 1 (HO-1) (catalog number: MBS8800478), cyclooxygenase 2 (COX-2) (catalog number: MBS7611878), IL-6 (catalog number: MBS370473), and nuclear factor-erythroid factor2-related factor 2 (Nrf-2) (catalog number: MBS012148) reagent kits were obtained as the rat ELISA kit from My Biosource, Inc. (San Diego, Southern California, CA, USA). All reagents were of analytical grade and commercially available.

### 3.2. Methods

#### 3.2.1. Preliminary Study 

The effect of several parameters such as sonication time, hydration medium, lipid content, and medication concentration on BSVC formation was investigated in the beginning. All factors were optimized in order to achieve the best BSVC size and entrapment. In actuality, BSVC vesicles were made with a combination of lecithin and cholesterol to increase vesicle entrapment and stability [95]. As an edge activator, SDC, SC and SGC were utilized in various concentrations. PBS (pH 7.4) as the hydration medium, chloroform-methanol mixture as the organic solvent, 36 mg cholesterol and lecithin (1:1 *w/w*), 10 mg DNa, 20 min sonication and 1 h hydration were all shown to be ideal for vesicle formation.

#### 3.2.2. Design and Optimization of Experiments

Design-Expert 10 software (Stat-Ease, Inc., Minneapolis, MI, USA) was used to design the most plausible 12 formulations according to multilevel categoric design 3^1^.2^2^. For optimization, three factors (independent variable): bile salts with three levels (SC, SDC, and SGC); surfactants with two levels (Span 40 (S40) and Span 60 (S60)); bile salts amount with two levels (8 and 18 mg) were provided to design expert software. Vesicle size (VS), polydispersity index (PDI), entrapment efficiency (EE%), Zeta potential (ZP), and the cumulative amount of DNa that permeated a rat skin specimen after 21 h (Q21) were chosen as the response variables. Table 7 and Table 2 clarify the levels of independent variables and the composition of the various experimental runs established utilizing the multilevel categoric design.

#### 3.2.3. Preparation of Diclofenac Sodium-Loaded Bilosomes (DNa-BSVCs)

A thin film hydration method was applied for the preparation of DNa-BSVCs [54]. Briefly, 50 mg of DNa, 180 mg of different span (Span 40 or Span 60) and 36 mg of cholesterol: lecithin (1:1) together with different weights of the bile salt used (SC or SGC or SDC) were precisely weighed and solubilized in round bottom flask with a mixture of chloroform–methanol (2:1). The organic solution was evaporated under decreased pressure at 40 °C in a rotary evaporator (Rotavapor, Heidolph VV 2000, Burladingen, Germany) until a thin dry film was formed. The lipid film was hydrated by adding phosphate buffer saline (PBS, 10 mL, pH 7.4) to the flask, which was rotated at 100 rpm for 1 h under normal pressure. The dispersion was sonicated for 20 min using an Ultrasonic bath sonicator (Model SH 150-41; Sonix TV ss-series, North Charleston, SC USA) [96]. Finally, the formulation was kept in a refrigerator (4 °C) until it was tested further. 

#### 3.2.4. Characterization of the Experimental Runs

##### Vesicle Size (VS), Polydispersity Index (PDI) and Surface Charge Analysis (ZP)

DNa-BSVCs vesicle size and surface charge analysis were assessed using Zetasizer Nano 7.11 (Malvern Instruments, Malvern, UK) [97]; 0.1 mL of each dispersion was mixed well with 9.9 mL of deionized water before the measurements to avoid the multi-scattering phenomena. All measurements were performed thrice at 25 °C and an angle of 90 °C to the incident light beam, and the mean values ± SD obtained were reported [98].

##### DNa Entrapment

The EE% of DNa in BSVC was calculated indirectly by subtracting the free DNa (non-entrapped drug) from the total amount of DNa incorporated in the formulation. Briefly, dispersions of DNa-BSVCs were centrifuged at 14,000 rpm for 2 h at 4 °C (Sigma Laborzentrifugen, Osterode am Harz, Germany). The yielded residue of DNa-loaded BSVCs was washed twice with PBS pH 7.4. Then, the clear supernatant was separated each time from BSVCs and filtered with a syringe filter (pore size: 0.45 nm nylon) (Millex-LG, Millipore Co., Billerica, MA, USA). At the end, the amount of free non- entrapped DNa was quantitatively assayed, in triplicate, spectrophotometrically using a UV spectrophotometer (Shimadzu UV-1800, Tokyo, Japan) at 276 nm [99,100]. The EE% of DNa was calculated using the following equation:(1)$$\%\mathrm{EE}=\frac{\mathrm{Total}\;\mathrm{drug}\;\mathrm{concentration}-\mathrm{free}\;\mathrm{drug}\;\mathrm{concentration}}{\mathrm{Total}\;\mathrm{drug}\;\mathrm{concentration}}\times 100$$

##### Ex Vivo Skin Permeation Study

To establish permeation of newborn rat skin, DNa-BSVCS formulations and DNa solution (control) were estimated using diffusion cell (Franz cell) with active permeation area (5.0 cm^2^) [101]. The recipient compartment was filled with a rat skin that had been carefully shaved, with the stratum corneum side facing the donor compartment. The recipient compartment was completed with 100 mL distilled water. Water was selected as permeation medium to simulate the in vivo sink conditions essential for permeation [102], and the donor compartment included DNa-BSVCs formulation (equivalent to 3 mg DNa) or DNa solution. The analysis was accompanied at 37 ± 2 °C with continuous stirring at 100 rpm for 21 h. At (1, 3, 6, 8, 9, 12, 15, 18 and 21 h) time intervals, samples were detached, and the DNa content was calculated using a spectrophotometer at λ_max_ of 276 nm. To maintain a sink state, the receiver compartment was replenished with new distilled water in an equal amount after each withdrawal. The permeated amounts of DNa from the rat skin per unit area (µg/cm^2^) were plotted versus time (h) for each BSVCs. The lag time (Lt) is the *x*-axis intercept by the linear portion of the graph. Permeation parameters, including the steady state flux (Jss) in µg/cm^2^/h, permeability coefficient (P) in cm^−2^/h^−1^ and enhancing factor (Fen) were determined for each DNa-BSVC formulation to attain the enhancement in the DNa permeation in comparison to the DNa solution (control) [55].

##### Selection of Optimized DNa-BSVC

The optimized formulation was selected including desirability obtained from Design-Expert 10 expert software [55]. The optimization procedure was considered in order to choose a formulation with the highest EE%, ZP and Q_21_, as well as the shortest vesicle size. The desirability index near to one was selected for the solution. To confirm the prediction ability, the optimized DNa-BSVC formulation was primed and characterized, and the actual values of EE%, ZP, Q_21_ h, and VS were paralleled with the predicted values.

#### 3.2.5. Optimized DNa-BSVC Characterization

##### In Vitro Release Evaluation

In vitro release tests used Franz diffusion cells (5 cm^2^ Surface area). A cellulose dialysis membrane (Mol. Wt. cut off = 12,000 Da) separated the donor chamber from the receptor chamber [103]. A certain volume of the optimum DNa-BSVC and DNa solution was kept in the donor chamber (equivalent to 3 mg) [104] injected 50 mL of PBS into the receptor chamber with steady stirring (50 rpm, 37 °C); 1 mL from the receptor compartment was withdrawn and replaced with 1 mL of fresh PBS at predetermined intervals. A “UV spectrophotometer” with a maximum wavelength of 276 nm was assessed to compute the DNa concentration of the aliquots.

##### Morphological Evaluation

Using TEM (JEM 1230, Joel, Tokyo, Japan), the surface properties of the determined optimum formulation were evaluated. A drop of undiluted dispersion was layered on a carbon-coated copper grid, allowed to adhere for about 1 min, and then allowed to dry at room temperature for 10 min. The air-dried sample was finally visualized at 20 and 25 KV [31].

##### Fourier Transform Infrared Spectroscopy (FTIR) of the Optimized DNa-BSVC

On an FT-IR spectrometer (IR435-U-04, Shimadzu, Kyoto, Japan) with an attenuated total reflectance cell in the range of 4000–400 cm^−1^, resolution 4 cm^−1^, FT-IR spectra of different samples were analyzed. For each material, three replicates were obtained, and spectra were further analyzed using the instrument’s software [105]. 

#### 3.2.6. Formulation of DNa-BSVC-Based Gel

According to a previously described approach, the improved DNa-BSVC formulation and free DNa solution were collective with Carbopol 971P polymer to generate a gel [106]. Carbopol 971P (2% *w/w*) and preservatives (0.01% propylparaben and 0.1% methylparaben) were sprinkled in water for 2 h while stirring. After that, DNa-BSVC was added to the gel basis (1% *w/w* of DNa) and stirred for 1 h. The pH was adjusted to 6.0 ± 0.05 using triethanolamine to produce a DNa-BSVC gel that was strong enough for topical use.

#### 3.2.7. Characterization of DNa-BSVC Based Gels

##### Rheological Characterization of the Hydrogel Formulations 

The rheological properties of the DNa hydrogel and DNa-BSVC hydrogel were measured using a rotational Brookfield viscometer of cone and plate structure (DV-III Ultra viscometer, RV model, Brookfield, USA). A sample (0.5 mL) of the formulation was added to the viscometer’s lower plate. The analysis was performed at 25 ± 1 °C using spindle 52 linked to the viscometer by a circulating bath at a shear rate varying from 10 to 200 (sec^−1^) [107]. The viscosities, as well as the area of hysteresis loops, were assessed. The area of hysteresis loops was calculated by trapezoidal rule, then deducting the area under the down curve from the area under the upper curve. To determine the flow behavior of dissimilar gel bases, Farrow’s equation was applied.
(2)Log G= N Log F–Log η
where G is the shear rate (sec^−1^), F is the shear stress (dyne/cm^2^), η is the viscosity (cp), and N is the constant of Farrow. In order to obtain the value of N that indicates the deviation from Newtonian law, log G was plotted versus log F. When N is less than one, dilatant flow (shear rate thickening) is suggested. If N is greater than one, plastic or pseudoplastic flow (shear rate thinning) is designated [108].

##### Ex Vivo Permeability Study 

Permeability studies of DNa-BSVC hydrogel formulation were performed and compared to DNa hydrogel as described above in the ex vivo permeability study using Franz diffusion cell.

#### 3.2.8. Animal Experiment

Male Wistar rats (220–250 g) were used in this study. The rats were housed in a sterile setting (temperature, relative humidity and lighting). All of the experiments were carried out with the approval of Cairo University’s Local Institutional Animal Ethics Committee (Acceptance No: 1721) and in accordance with the United States National Academy of Sciences’ Guide for the Care and Use of Laboratory Animals, which was published in 2011.

#### Histopathological Investigation of the DNa-BSVC-Based Gel

The histopathological examination was carried out to determine the safety of using the DNa-BSVC gel topically. Six rats were divided into two groups at random. Group A acted as a control, while the other group received the optimized DNa-BSVC gel topically on the hairless dorsal skin surface for seven days. The rats were then killed, and their skin was taken for histological analysis. For 24 h, the removed skin was preserved in 10% formaldehyde. The parts were cleaned in xylene and then placed in paraffin wax blocks for 24 h at 56 degrees Celsius. The sections were cut into five-millimeter thick slices, then mounted on glass slides and stained with hematoxylin and eosin (H&E) [109]. Finally, stained slices for topical usage were examined under a light microscope.

#### 3.2.9. In Vivo Study of DNa-BSVCs Based Gel

##### Animals

Adult male Wistar rats (average weight 190 g) were used in the present investigation. Animals were obtained from an animal house of Cairo University, Egypt. Animals were reserved under observation for about two weeks before starting the experimental protocol to avoid any inter-current contamination at an animal house of Cairo University, Egypt. The chosen animals were housed in experimental cages, at a constant temperature (25 ± 1 °C) and light/dark (12/12 h) cycles. Free access to water and standard forage ad libitum were allowed for animals. All animal guide rules were performed according to the Cairo University guidelines and complied with the research protocols applied by the Animal Care Committee of the National Research Center (Cairo, Egypt) which comply with the recommendations of the National Institutes of Health (NIH) guide for the care and use of experimental animals.

##### Experimental Design

Rats were randomly allocated into 4 weight-matched groups, each of 10 rats. The first group was kept as a normal control group and received only vehicles. The second group was kept as a positive control group and received only Carrageenan (0.1 mL of 1.5%) [106,110]. The third one was kept as the DNa-based gel group, and the fourth was kept as the DNa-BSVC based gel group with gel applied at a dose (10 mg/kg body mass) [111]. Group 3 and 4 agents were applied after exposure to induction with 30 min on the rats’ dorsal region to allow systemic drug influence. Doses of tested groups were determined with pilot trials guided by the published research.

##### Induction of Rat Paw Edema

Induction of paw edema was performed by injecting rats with a single dose of carrageenan at each rat’s right hind paw intraplantar at a dose (0.1 mL) of 1.5% prepared before induction 24 h by dissolving in saline and stored 2–4 °C [106,110].

##### Measurement of Rat Paw Volume

Rat paw volume % inhibition was assessed starting from time zero and respectively for 5 h after induction with carrageenan using a plethysmometer according to the method carried out by Newbould [112].

##### Serum Sampling

At the end of the test, blood was collected from the retro-orbital plexus under light ether anesthesia using heparinized tubes. Blood was left to coagulate on crushed ice then centrifuged under the relative centrifugal force of 1000× *g* for 10 min using a cooling centrifuge (Sigma Bench Top cooling centrifuge, Model 2-16kl; Montreal Biotech Inc; Germany). The clear serum layer was carefully withdrawn and stored in a deep freezer at −80 °C until the time of assay of serum HO-1 [113] and Nrf-2 [114]. 

##### Tissue Sampling

Animals were sacrificed by cervical dislocation, right and left paws were punched. One of them was preserved at −80 °C until the time of assay of IL-6 and COX-2 [115,116]; the second was fixed in 10% isotonic formalin solution in normal saline for at least 48 h until the time of the histopathological study.

##### Biomarkers Estimated Using ELISA Technique

COX-2, IL-6, Nrf-2 and HO-1 were estimated using ELISA test reagent kits in compliance with the manufacturer’s instructions at 450 nm. These biomarkers were measured with the aid of an ELISA Processing System (Spectra Max Plus-384 Model Absorbance Microplate Reader, San Jose, CA, USA). 

#### 3.2.10. Statistical Analysis

The significant difference among group means was conducted via one-way ANOVA, and then a Tukey–Kramer multiple comparisons test was applied, with the aid of Graph Pad Instat computer software (Software Graph Pad, San Diego, CA, USA). Added to that, Microsoft Excel 2010 software (Microsoft Corporation, Microsoft Redmond campus in Redmond, WA, USA) was handled to plot graphs and tables.

## 4. Conclusions

The conducted study revealed that a new novel DNa-incorporated BSVCs hydrogel is an effective transdermal carrier for the treatment of local and systemic inflammation experimentally induced in a rat model. The formulation variables were statistically optimized using a multilevel categoric design. DNa-BSVCs exhibited a high EE percentage, a small particle size, a reasonable zeta potential, and a 21 h ex vivo permeation profile. The optimized DNa-BSVCs demonstrated a non-aggregating spherical structure and prolonged in vitro release profile over 8 h. S12 was then incorporated into Carbopol 971P gel. A rheological examination and ex vivo analysis of the prepared gels revealed more desirable rheological parameters and a higher skin permeation, so it was chosen for further in vivo characterization. DNa-BSVCs hydrogel was found to best fit to inhibit all signs of inflammation represented via carrageenan in the experimental animals. DNa-BSVCs hydrogel exhibited a significant reduction in the rat paw volume % inhibition, restoring it to normal, representing potential local effects for DNa-BSVCs hydrogel in reducing swelling, redness and pain. In addition, the systemic tissue levels of COX 2 and IL-6 revealed a significant down-regulation compared to the carrageenan group; however, Nrf 2 and HO-1 serum levels showed a significant upregulation. Furthermore, histopathological examination showed normal bone structure coupled with normal surrounding soft tissues free from edema and inflammation. Finally, our research presented a superior anti-inflammatory activity of DNa-BSVCs hydrogel over the DNa plain gel in inhibiting signs of inflammation coupled with omitting DNa oral application side effect. However, further clinical trials are supposed to confirm such findings.

## Figures and Tables

**Figure 1 pharmaceuticals-15-01106-f001:**
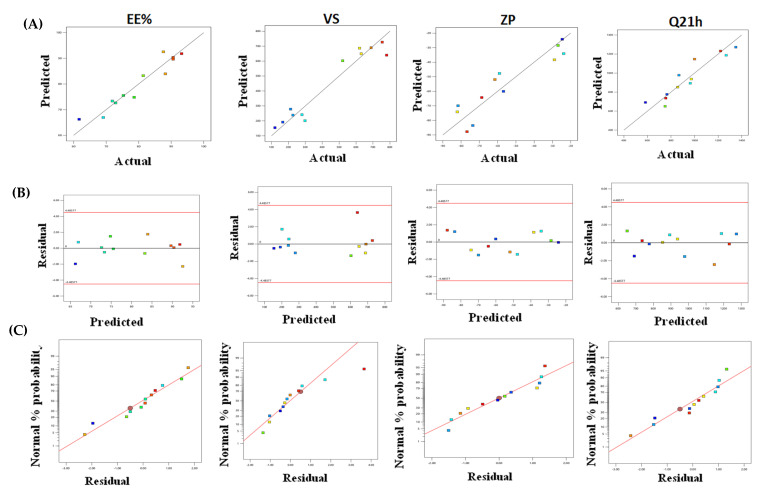
Model diagnostic plots of the three independent variables (**A**) linear correlation plots between actual and predicted values for various responses of DNa-BSVC, (**B**) plot of residual error vs. model predicted responses and (**C**) normal quantile-quantile plots of residual errors.

**Figure 2 pharmaceuticals-15-01106-f002:**
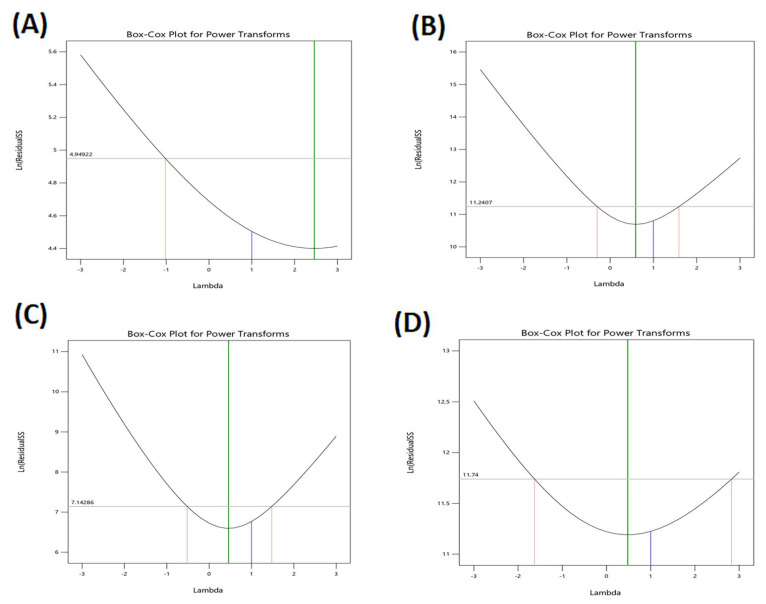
Box–Cox plot of all responses. EE: entrapment efficiency (**A**); VS: vesicle size (**B**); ZP: zeta potential (**C**); Q21 h: cumulative amount of DNa permeation (**D**).

**Figure 3 pharmaceuticals-15-01106-f003:**
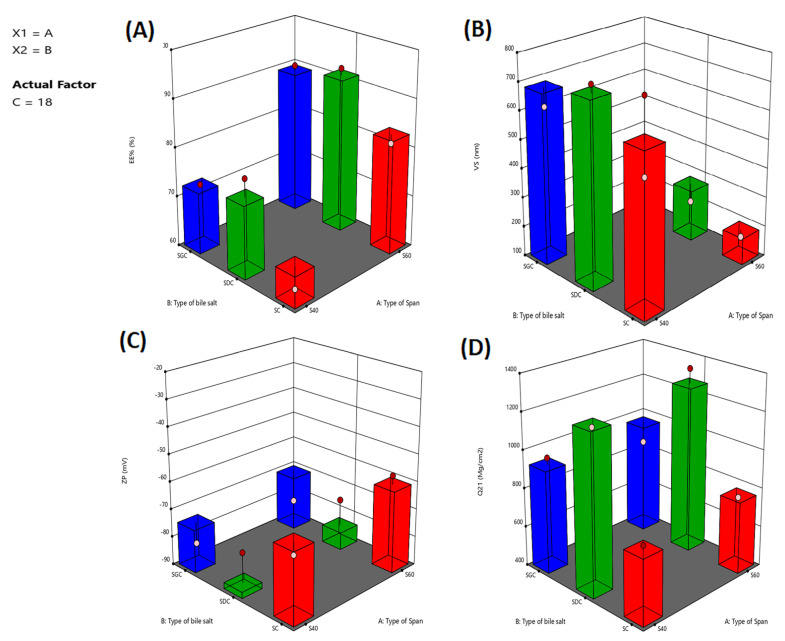
3D response surface plots showing the influence of span type, bile salt type and bile salt amount of DNa-BSVCs on different responses of EE: entrapment efficiency (**A**); VS: vesicle size (**B**); ZP: zeta potential (**C**); Q21 h: cumulative amount of drug permeated (**D**).

**Figure 4 pharmaceuticals-15-01106-f004:**
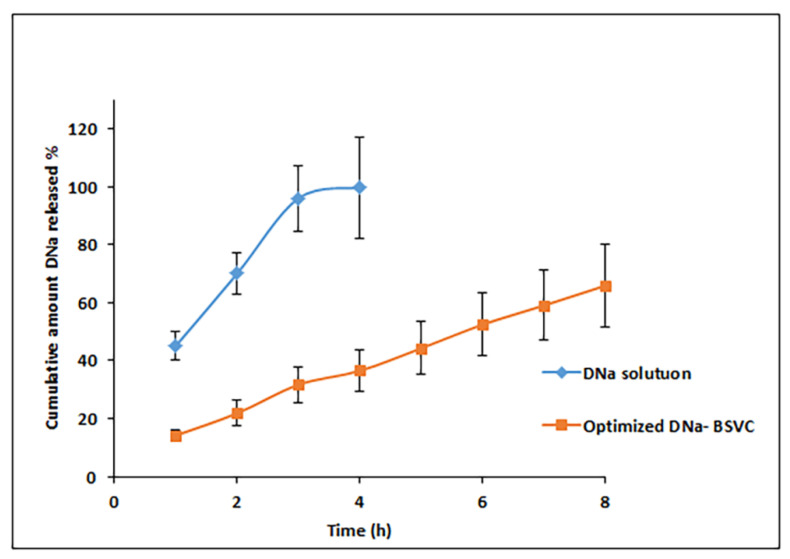
In vitro release profile of DNa from BSVC (optimal formulation) and drug solution.

**Figure 5 pharmaceuticals-15-01106-f005:**
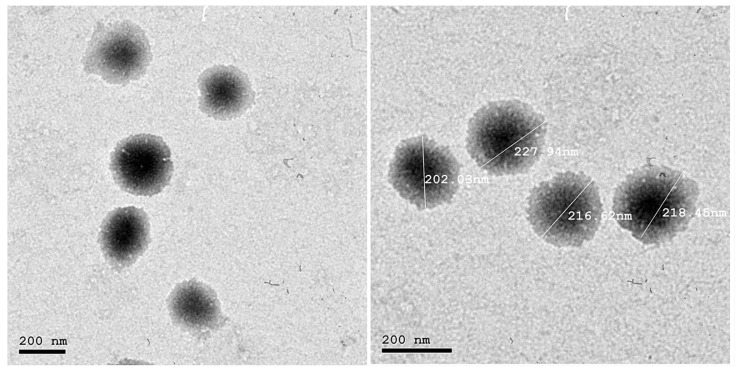
Transmission electron microscopy micrograph of the optimized DNa-BSVC formulation.

**Figure 6 pharmaceuticals-15-01106-f006:**
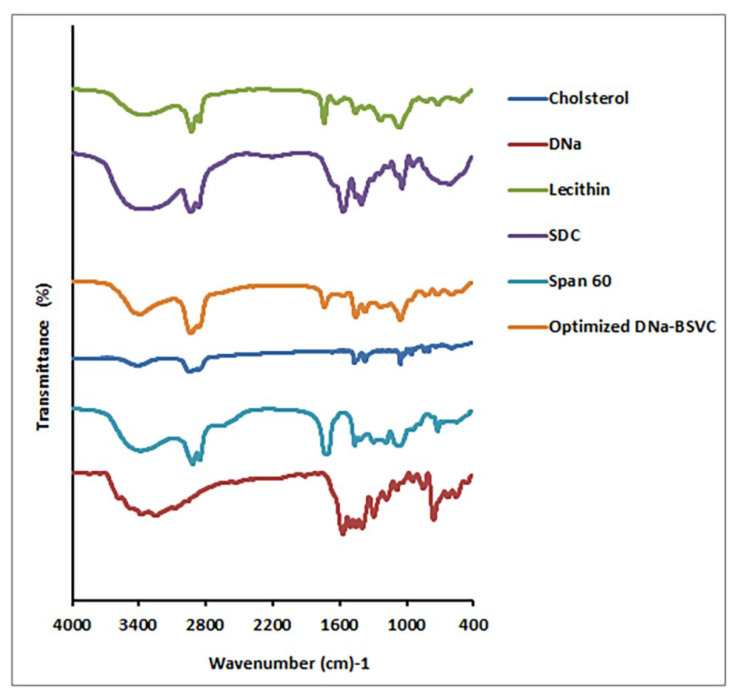
FTIR spectra of DNa, lecithin, cholesterol, SDC, Span 60 and optimized BSVC formulation.

**Figure 7 pharmaceuticals-15-01106-f007:**
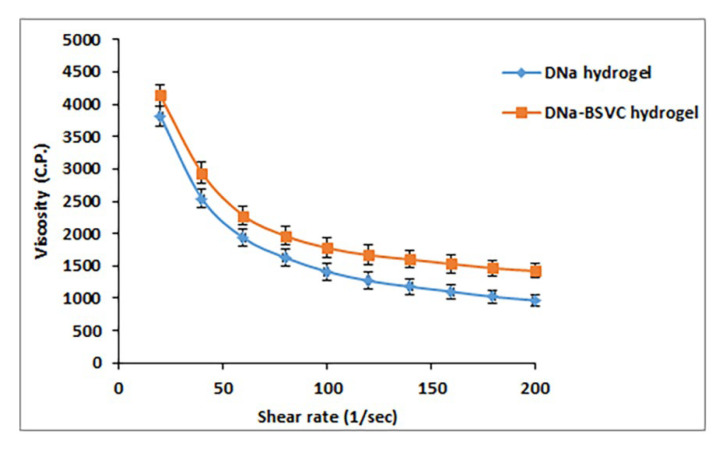
Rheology of DNa hydrogel and DNa-BSVC hydrogel formulations at 25 ± 1 °C (*n* = 3 ± SD).

**Figure 8 pharmaceuticals-15-01106-f008:**
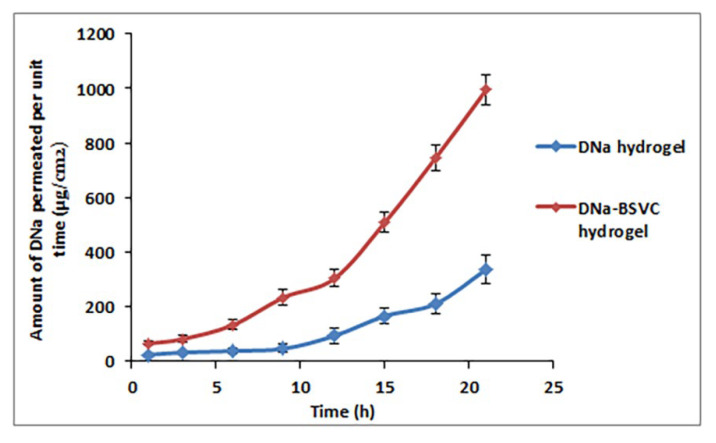
Ex vivo skin permeation of DNa hydrogel and DNa-BSVC hydrogel.

**Figure 9 pharmaceuticals-15-01106-f009:**
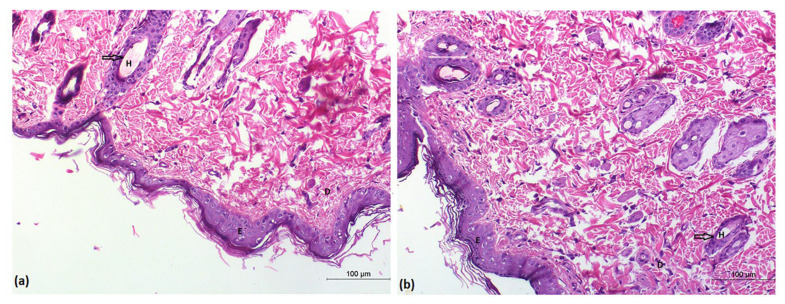
Representative histopathological images of untreated rat skin group (**a**); and DNa-BSVC gel treated rat skin group (**b**).

**Figure 10 pharmaceuticals-15-01106-f010:**
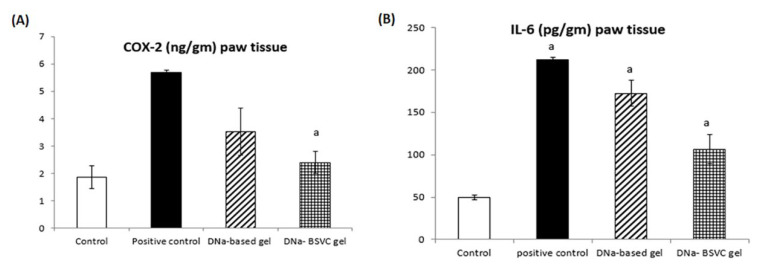
Treatment effects of DNa-based gel and DNa-BSVC gel on COX-2 in paw tissue homogenate of CA-induced paw edema in rats (**A**) and IL-6 (in paw tissue homogenate of CA-induced paw edema in rats (**B**). Statistical analysis and the significance of difference between groups’ means were performed via one-way ANOVA, and after that a Tukey–Kramer multiple comparisons test at *p* > 0.05. The values are expressed as inhibitory rate (%) for 6 rats data identified as mean ± SEM. a: significantly different from normal control.

**Figure 11 pharmaceuticals-15-01106-f011:**
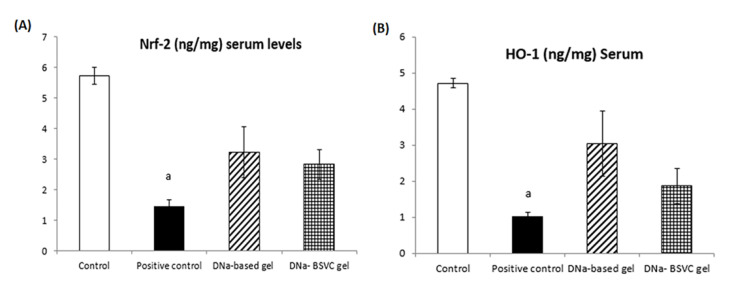
Treatment effects of DNa-based gel and DNa-BSVC gel on Nrf-2 in serum levels against CA-induced paw edema in rats (**A**); and HO-1 in serum levels against CA-induced paw edema in rats (**B**). Statistical analysis and the significance of difference between groups’ means were performed via one-way ANOVA, and after that a Tukey–Kramer multiple comparisons test at *p* > 0.05. The values are expressed as inhibitory rate (%) for 6 rats; data identified as mean ± SEM. a: significantly different from Normal control.

**Figure 12 pharmaceuticals-15-01106-f012:**
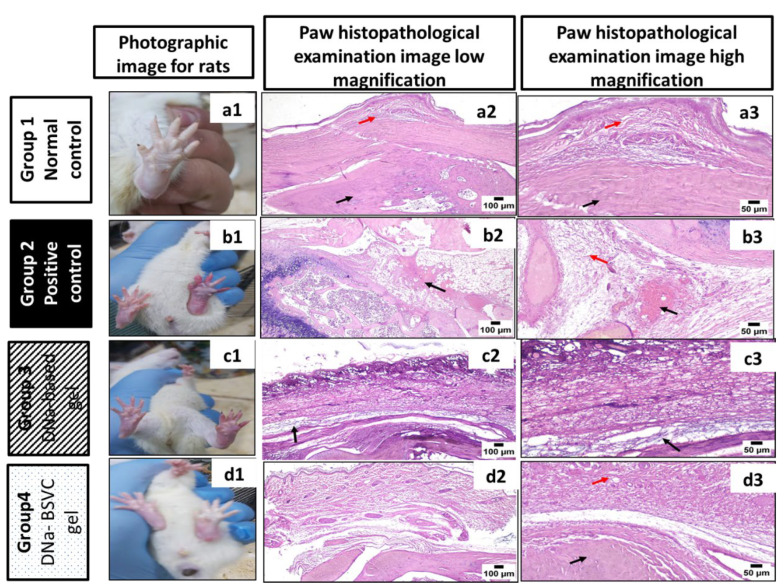
Photographic image for rats and photomicrographs of histopathological examination of paws (100 μm and 50 μm magnification) of each group. Where: (**a1**–**a3**), normal control group; (**b1**–**b3**), positive control group (induction with carrageenan); (**c1**–**c3**), DNa-based gel group (10 mg/kg); (**d1**–**d3**), DNa-BSVC gel group (10 mg/kg); (Figure **a2,a3**): photomicrograph of paw, control group (Figure **b3**): group showing marked edema (red arrow) and hemorrhage (black arrow) in the periarticular tissue, DNa-based gel; (Figure **c2**,**c3**) showing moderate edema (black arrow) in the tissue surrounding the bone, and a DNa-BSVC gel group; (Figure **d2**,**d3**) showing normal structure of bone (black arrow) and surrounding soft tissues (red arrow) (H&E).

**Table 1 pharmaceuticals-15-01106-t001:** Analysis of variance of the measured responses data.

Source	EE%		VS		ZP		Q_21_	
	F	*p* Value	F	*p* Value	F	*p* Value	F	*p* Value
Model	19.93	0.0006	22.17	0.0004	10.12	0.0049	12.25	0.0028
X1 = A = Span type	67.28	<0.0001	85.93	<0.0001	0.4340	0.5311	0.4599	0.5195
X2 = B = Bile salt type	6.16	0.0287	1.08	0.3892	4.49	0.0557	23.25	0.0008
X3 = C = Bile salt amount	0.12	0.7399	0.59	0.4670	31.09	0.0008	2.03	0.1973
Adjusted R^2^	0.8731		0.8850		0.7684		0.8038	
R^2^	0.9193		0.9268		0.8526		0.8750	
%CV	4.49		18.97		20.10		10.94	
Predicted R^2^	0.7622		0.7850		0.5669		0.6326	
Adequate precision	11.3421		10.59		8.8439		9.3077	
SD	3.59		83.78		11.13		103.48	
EE %=+80.09+8.51A−4.99B (1)+3.58B (2)−0.3585C								
VS=+441.68−224.18A−44.66B (1)+42.44B (2)+18.60C								
ZP=−55.37+0.0667A+11.12B (1)−12.36B (2)−17.92 C								
Q21 h=+945.94+2.12A−232.54B (1)+263.56B (2)+42.56C								

VS: vesicles sizes; EE%: entrapment efficiency percent; ZP: zeta potential; Q21: cumulative amount permeated/unit area in 21 h; F: Fisher’s ratio (F = MS Factor/MS Residual); CV: Percentage coefficient of variation (calculated as SD/mean * 100); SD: standard deviation. Data are mean values (*n* = 3) ± SD.

**Table 2 pharmaceuticals-15-01106-t002:** The 3^1^2^2^ multilevel categoric design and the observed responses of DNa-BSVCs.

Formulation	Dependent Variables							
	A: Span Type	B: Bile Salt Type	C: Bile Salt Amount (mg)	Y_1_: EE%	Y_2_: Vesicle Size (nm)	Y_3_: ZP (mV)	Y_4_: Q_21_ (µg/cm^2^)	PDI
S1	S40	SDC	8	75.30 ± 7.76	688.9 ± 9.24	−61.5 ± 10.32	999.6 ± 26.02	0.47
S2	S40	SC	8	69.1 ± 10.66	520.6 ± 8.91	−26.90 ± 4.20	749.8 ± 23.35	0.37
S3	S60	SDC	8	87.6 ± 11.23	280.2 ± 0.78	−59.00 ± 9.45	1268.1 ± 38.25	0.35
S4	S40	SGC	18	72.9 ± 8.96	620.6 ± 5.67	−82.1 ± 12.63	972.6 ± 33.87	0.40
S5	S60	SC	18	81.4 ± 0.56	166.7 ± 2.56	−56.80 ± 7.14	765.3 ± 32.60	0.32
S6	S60	SGC	18	90.6 ± 6.44	226.6 ± 7.81	−81.7 ± 11.23	867.2 ± 32.77	0.36
S7	S60	SC	8	88.2 ± 9.52	120.4 ± 0.65	−24.60 ± 3.47	582.9 ± 32.14	0.31
S8	S40	SGC	8	71.9 ± 5.31	630.2 ± 3.56	−28.90 ± 4.65	856.4 ± 35.33	0.43
S9	S60	SGC	8	90.6 ± 1.45	298.2 ± 4.67	−23.80 ± 2.65	963.5 ± 37.52	0.38
S10	S40	SC	18	61.7 ± 3.44	780.4 ± 0.99	−68.70 ± 8.65	755.6 ± 30.21	0.62
S11	S40	SDC	18	78.6 ± 2.37	754.5 ± 6.22	−76.80 ± 6.89	1220.1 ± 43.89	0.49
S12	S60	SDC	18	93.2 ± 2.21	212.9 ± 0.56	−73.6 ± 13.24	1350.2 ± 45.41	0.33
Optimized formulation	S60	SDC	18					

S40: Span 40; S60: Span 60; SC: sodium cholate; SGC: sodium glucocholate; SDC: sodium deoxycholate; EE%: entrapment efficiency percent; VS: vesicles sizes; ZP: zeta potential; Q21: cumulative amount permeated/unit area in 21 h; PDI: polydespersity index. Data are mean values (*n* = 3) ± SD.

**Table 3 pharmaceuticals-15-01106-t003:** Ex vivo permeation parameters of DNa-BSVCs versus DNa solution.

Formulation	Lag Time (h)	Jss (µg/cm^2^ h)	Kp (cm/h)	EI
S1	4.51 ± 0.068	58.87 ± 0.67	0.0196 ± 0.005	2.74 ± 0.467
S2	5.37 ± 0.332	48.11 ± 1.22	0.0160 ± 0.001	2.06 ± 0.662
S3	4.97 ± 0.121	71.64 ± 0.45	0.0238 ± 0.007	3.48 ± 0.661
S4	6.17 ± 0.235	63.51 ± 0.36	0.0652 ± 0.007	2.67 ± 0.056
S5	5.15 ± 0.213	49.00 ± 0.56	0.0163 ± 0.005	2.10 ± 0.324
S6	3.63 ± 0.346	50.69 ± 0.89	0.0168 ± 0.001	2.38 ± 0.789
S7	2.42 ± 0.091	31.97 ± 0.98	0.0106 ± 0.004	1.60 ± 0.567
S8	4.01 ± 0.042	50.39 ± 0.66	0.0167 ± 0.008	2.35 ± 0.789
S9	6.52 ± 0.245	65.53 ± 0.78	0.0218 ± 0.009	2.64 ± 1.45
S10	1.52 ± 0.056	41.53 ± 1.64	0.0138 ± 0.005	2.07 ± 2.67
S11	2.71 ± 0.049	65.33 ± 0.452	0.0217 ± 0.003	3.35 ± 0.185
S12	6.23 ± 0.032	87.24 ± 0. 356	0.0290 ± 0.006	3.70 ± 0.861
Drug solution	6.17 ± 0.345	23.44 ± 0.743	0.0078 ± 0.003	-

Jss: drug flux; Kp: permeability coefficient; EI: enhancement index. Data are mean values (*n* = 3) ± SD.

**Table 4 pharmaceuticals-15-01106-t004:** Experimental, model expected and prediction error values of the optimized DNa-BSVC formulation.

Response Variables	Experimental Value	Expected Value	Prediction Error (%) *
EE%	95.41%,	91.82%	3.76
VS	275.33 nm	278.542 nm	1.17
ZP	−86.32 mV	−83.53 mV	3.23
Q_21_	1273.85 µg/cm^2^	1241.92 µg/cm^2^	2.51

* Calculated as (experimental model expected)/experimental × 100.

**Table 5 pharmaceuticals-15-01106-t005:** Rheological parameters of the prepared DNa hydrogel and DNa-BSVC hydrogel.

Formulation	At 37 °C		Farrow’s Constant N	Flow Behavior	Area of Hysteresis Loop (Dyne/cm^2^.sec)
	Viscosity (Min) (cp)	Viscosity (Max) (cp)			
DNa hydrogel	970 ± 83.2	3816 ± 152.7	2.48 ± 0.07	Shear rate thinning with thixotropy	1224.60 ± 102.56
DNa-BSVC hydrogel	1423 ± 113.4	4135 ± 172.2	1.85 ± 0.04	Shear rate thinning with thixotropy	1956.19 ± 134.85

**Table 6 pharmaceuticals-15-01106-t006:** Effect of DNa-BSVC gel on % increase paw edema volume measured for 5 h period prior to rats being subjected to carrageenan injection.

Response Variables	Normal Control (Inhibit. %)	P.C (CA 1%) (Inhibit. %)	DNa- Based Gel (Inhibit. %)	DNa-BSVC Gel (Inhibit. %)
Zero time	0.99 ± 0.014	0.85 ± 0.150	0.85 ± 0.150	0.99 ± 0.014
1 h	1.49 ± 0.008 (50.9%) ^(a,b,c)^	1.75 ± 0.016 (11.3%) ^(a,b)^	1.75 ± 0.016 (11.3%) ^(a,b)^	1.49 ± 0.008 (50.9%) ^(a,b,c)^
2 h	1.40 ± 0.004 (59.6%) ^(a,b,c)^	1.69 ± 0.018 (15.3%) ^(a,b)^	1.69 ± 0.018 (15.3%) ^(a,b)^	1.40 ± 0.004 (59.6%) ^(a,b,c)^
3 h	1.34 ± 0.008 (65.2%) ^(a,b,c)^	1.64 ± 0.010 (20.3%) ^(a,b)^	1.64 ± 0.010 (20.3%) ^(a,b)^	1.34 ± 0.008 (65.2%) ^(a,b,c)^
4 h	1.31 ± 0.006 (67.9%) ^(a,b,c)^	1.61 ± 0.008 (22.7%) ^(a,b)^	1.61 ± 0.008 (22.7%) ^(a,b)^	1.31 ± 0.006 (67.9%) ^(a,b,c)^
5 h	1.29 ± 0.005 (69.1%) ^(a,b,c)^	1.60 ± 0.007 (23.2%) ^(a,b)^	1.60 ± 0.007 (23.2%) ^(a,b)^	1.29 ± 0.005 (69.1%) ^(a,b,c)^

Statistical analysis and the significance of difference between groups’ means were performed via one-way ANOVA, and after that, a Tukey–Kramer multiple comparisons test at *p* > 0.05. The values are expressed as inhibitory rate (%) for 6 rats data identified as mean ± SEM. ^a^: significantly different from normal control; ^b^: significantly different from positive control; ^c^: significantly different from DNa-based gel.

**Table 7 pharmaceuticals-15-01106-t007:** Variables and their corresponding levels in the employed 3^1^2^2^ multilevel categoric design for DNa-loaded BSVCs.

Variable	Design Level		
	Low (−1)	Medium (0)	High (+1)
Independent variables			
A = Span type	Span 40		Span 60
B = Bile salt type	SC	SDC	SGC
C = Bile salt amount (mg)	8		18

SC: sodium cholate; SGC: sodium glucocholate; SDC: sodium deoxycholate. Data are mean values (*n* = 3) ± SD.

## Data Availability

Data is contained within the article and Appendix A.

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
