# Peer review of "Novel Bile Salt Stabilized Vesicles-Mediated Effective Topical Delivery of Diclofenac Sodium: A New Therapeutic Approach for Pain and Inflammation"

_pharmaceuticals, 2022, doi:10.3390/ph15091106_

Round 1
Reviewer 1 Report
For ex vivo study, samples were analyzed by spectrophotometer; what about the chemicals leaching from skin which also absorb at 276 nm? HPLC assay should have been used.
Why rat skin used for ex vivo study; it is not a good model for human skin. Pig skin or human skin should be used. Why water in receptor rather than pH7.4 PBS?
Why do we expect vesicles to increase permeation? Is it just the enhancers in the vesicles?
Wording is very poor in several places, for example, estimate the prospective, In addition to,....., publicized, "there is currently, no evidence was..", water was elected as permeation milieu, animal was killed (should be sacrificed),preponderence of, needles to say (even needless to say is not good wording) etc.
In conclusion, treatment of carrageenan-induced rat paw edema is not a benefit!
Author Response
Point 1: For ex vivo study, samples were analyzed by spectrophotometer; what about the chemicals leaching from skin which also absorb at 276 nm? HPLC assay should have been used.
Response 1: ex vivo study discussed according to ref. (Mallina, S.A, et al, 2018).
Mallina, S.A. and Sundararajan, R., 2018. Diclofenac sodium loaded liposomal gel for transdermal delivery: Formulation, characterisation and pharmacokinetic evaluation. Research Journal of Pharmacy and Technology, 11(9), pp.4051-4062.
The drug only absorbed at 276 nm and there are many reference validated our study as (Akbari, J. et al, 2022)
Akbari, J., Saeedi, M., Morteza-Semnani, K., Hashemi, S.M.H., Babaei, A., Eghbali, M., Mohammadi, M., Rostamkalaei, S.S., Asare-Addo, K. and Nokhodchi, A., 2022. Innovative topical niosomal gel formulation containing diclofenac sodium (niofenac). Journal of Drug Targeting, 30(1), pp.108-117.
POINT 2: Why rat skin used for ex vivo study; it is not a good model for human skin. Pig skin or human skin should be used. Why water in receptor rather than pH7.4 PBS?
Response 2: Rat skin used normally for ex vivo permeation and we discussed according to ref. (Tran, Y.T.H et al, 2020)
Tran, Y.T.H., Tran, G.N., Hoang, A.L. and Vu, G.T.T., 2020. Niosomes loaded with diclofenac for transdermal administration: Physico-chemical characterization, ex vivo and in vivo skin permeation studies. Journal of applied pharmaceutical science, 10(12), pp.053-061.
We used water to simulate sink condition and we discussed according to ref. (Tavano, L, et al 2018)
Tavano, L., Mazzotta, E. and Muzzalupo, R., 2018. Innovative topical formulations from diclofenac sodium used as surfadrug: The birth of Diclosomes. Colloids and Surfaces B: Biointerfaces, 164, pp.177-184.
Point 3: Why do we expect vesicles to increase permeation? Is it just the enhancers in the vesicles?
Response 3: yes there are enhancers in this formulation as sodium cholate, sodium deoxycholate and sodium glycocholate
Point 4: Wording is very poor in several places, for example, estimate the prospective, In addition to,....., publicized, "there is currently, no evidence was..", water was elected as permeation milieu, animal was killed (should be sacrificed), preponderence of, needless to say (even needless to say is not good wording) etc.
Response 4: all these words are changed
Point 5: In conclusion, treatment of carrageenan-induced rat paw edema is not a benefit!
Response 5: This sentence is changed
Reviewer 2 Report
The manuscript is aimed to propose an innovative therapeutic approach for pain and inflammation based on bile salt stabilized vesicles-mediated effective topical delivery of diclofenac sodium.
The topic is appropriate for the journal.
The title is adequate and correlate with the content of the article.
The abstract reports a consistent summary of the article findings.
The work has a clear structure.
All sections are required for a complete understanding.
Nevertheless, there are minor issues that require to be addressed before proceeding with the publication, to enhance the quality and presentation to a broad audience.
· The whole manuscript would strongly benefit an English editing.
· Check for typos.
· It is suggested to add on a clear sentence stating the aim and hypothesis of the work at the very end of the introduction section.
· References are appropriately mentioned, but it worths mentioning that the paper would benefit more complete overview of a broader plethora of attempted approaches/materials in order to emphasize the scientific soundness of the presented findings (e.g., J. Biomed. Mater. Res. A, 104A, 2016, 1668-1679 : it provides for a background about the molecular mechanisms of lipids substitutives and their role within the pharmaceutics development.).
· The conclusion section would definitely benefit futher explanation, e.g. addition of a few sentences recapitulating the whole findings, the scientific progress and soundness of the original research work. It might help discuss over an additional section any possible limitations & perspectives.
· No TOC graphic is presented.
Author Response
Point 1: It is suggested to add on a clear sentence stating the aim and hypothesis of the work at the very end of the introduction section.
Response1: This clear sentence added at the very end of the introduction section.
Point 2: References are appropriately mentioned, but it worths mentioning that the paper would benefit more complete overview of a broader plethora of attempted approaches/materials in order to emphasize the scientific soundness of the presented findings (e.g., J. Biomed. Mater. Res. A, 104A, 2016, 1668-1679: it provides for a background about the molecular mechanisms of lipids substitutive and their role within the pharmaceutics development.).
Response 2: more reference cited to give complete overview of a broader plethora of attempted approaches/materials in order to emphasize the scientific soundness of the presented findings.
I could not find this reference (J. Biomed. Mater. Res. A, 104A, 2016, 1668-1679)
Point 3: The conclusion section would definitely benefit further explanation, e.g. addition of a few sentences recapitulating the whole findings, the scientific progress and soundness of the original research work. It might help discuss over an additional section any possible limitations & perspectives.
Response 3: More sentences clarified the benefited role of BSVCs- DNa added in conclusion
Point 4: No TOC graphic is presented.
Response 4: all figure adjusted
Round 2
Reviewer 1 Report
Carrageenin induced rat paw edema demonstrated delivery but this is not a benefit in itself; please revise conclusion as mentioned earlier
Author Response
Novel Bile salt stabilized vesicles- mediated effective topical delivery of Diclofenac Sodium: a new therapeutic approach for pain and inflammation
Tamer M. Mahmoud1, Mohamed M. Nafady2, Hanan O. Farouk2*, Dina M. Mahmoud2, Yasmin M. Ahmed3, Randa Mohammed Zaki4, 5, Doaa S. Hamad2
1 Department of Pharmaceutics, Faculty of Pharmacy, MTI University, Egypt; tamernafady@yahoo.com
2 Department of Pharmaceutics, Faculty of Pharmacy, Nahda University Beni-Suef, Egypt; mohamednafadynub@gmail.com (M.M.N); hanan.osman@nub.edu.eg (H.O.F); dnafady@yahoo.com ( D.M.M); doaa.saad@nub.edu.eg (D.S.A)
3 Department of Pharmacology and Toxicology, Faculty of Pharmacy, Nahda University Beni-Suef, Egypt; yasmain.mostafa@nub.edu.eg
4 Department of Pharmaceutics, college of Pharmacy, Prince Sattam Bin Abdulaziz University, Al-Kharj 11942, Saudi Arabia; randazaki439@yahoo.com
5 Department of Pharmaceutics and Industrial Pharmacy, Faculty of Pharmacy, Beni-Suef University, Beni-Suef 62514, Egypt; randazaki439@yahoo.com
* Correspondence: hanan.osman@nub.edu.eg; Tel.: (01005466740; 62511)
Reviewer Comment: Carrageenin-induced rat paw edema demonstrated delivery, but this is not a benefit in itself; please revise the conclusion as mentioned earlier
Response
This is the first study to use bilosomes as nano carrier for diclofenac sodium in the treatment of rat paw edema. Injection of a carrageenan solution into the pouch produces an inflammatory reaction that is characterized by an infiltration of cells, an increase in exudate, and a marked production of pro-inflammatory mediators, such as prostaglandins, leukotrienes, and cytokines, as well as components of the oxidative stress response which represents via a significant elevation in COX-2 and IL-6 and reduction in serum Nrf-2 and HO-1 to coupled with local inflammation and histopathological changes. In treatment with DNa-BSVCs hydrogel the systemic inflammation represents in both the tissue and serum significantly restored to normal compared to caragennane group., a; also histopathological study showed bone restoration without any signs of inflammation all are coupled with local effect represents via decreasing local edema and congestion after hydrogel application. These results presented a superior anti-inflammatory activity of DNa-BSVCs hydrogel over the DNa plain gel in inhibiting signs of inflammation and omitting DNa oral application side effects. However, further clinical trials are supposed to confirm such findings. This section added in conclusion to show the benefit of our study.